# QUICKDROP: EFFICIENT FEDERATED UNLEARNING BY INTEGRATED DATASET DISTILLATION

## ABSTRACT

Federated Unlearning (FU) aims to delete specific training data from an ML model trained using Federated Learning (FL). We introduce QUICKDROP, an efficient and original FU method that utilizes dataset distillation (DD) to accelerate unlearning and drastically reduces computational overhead compared to existing approaches. In QUICKDROP, each client uses DD to generate a compact dataset representative of the original training dataset, called a *distilled dataset*, and uses this compact dataset during unlearning. To unlearn specific knowledge from the global model, QUICKDROP has clients execute Stochastic Gradient Ascent with samples from the distilled datasets, thus significantly reducing computational overhead compared to conventional FU methods. We further increase the efficiency of QUICKDROP by ingeniously integrating DD into the FL training process. By reusing the gradient updates produced during FL training for DD, the overhead of creating distilled datasets becomes close to negligible. Evaluations on three standard datasets show that, with comparable accuracy guarantees, QUICKDROP reduces the duration of unlearning by $463.8\times$ compared to model retraining from scratch and $65.1\times$ compared to existing FU approaches. We also demonstrate the scalability of QUICKDROP with 100 clients and show its effectiveness while handling multiple unlearning operations.

## 1 INTRODUCTION

The vast amount of data produced by computing devices is increasingly being used to train large-scale ML models that empower industrial processes and personal experiences (Mahdavinejad et al., 2018). However, this data is often privacy sensitive or very large in volume, making it prohibitively expensive or impossible to upload it to a central server (Yang et al., 2019). To sidestep this issue, federated learning (FL) is increasingly being applied to collaboratively train ML models in a privacy-preserving manner (McMahan et al., 2017). FL obviates the need to move the data to a central location by having participants only exchange model updates with a server. In each round, this server aggregates all incoming trained models and sends the updated global model to participants.

Recent privacy regulations like the General Data Protection Regulation (GDPR) grant data owners with the "right to be forgotten" (European Union, 2018). In the realm of ML, this requires organizations to remove the influence of this data on the trained model (Zhang et al., 2023). The latter is called *machine unlearning* (Bourtoule et al., 2021). For instance, hospitals that collaboratively trained a model using FL might have to unlearn particular data samples in response to patient requests. However, the distributed nature of FL and the inability to access training data directly makes federated unlearning (FU) a challenging task.

A naive way to unlearn particular samples is to retrain the model from scratch while omitting these samples. As the model and data size increase, complete retraining quickly becomes prohibitively expensive regarding the time and compute resources required. A more effective approach is to use stochastic gradient ascent (SGA) on the samples being unlearned, *i.e.*, optimizing in the direction that maximizes the loss function (Wu et al., 2022). However, since SGA updates the entire model, it also deteriorates the model performance on the remaining samples. Hence, this approach typically utilizes a recovery phase, which retrains on the remaining samples for a few rounds to restore performance. While this method is more efficient than full retraining, the unlearning and recovery phases are still computationally demanding when dealing with a large volume of training data.

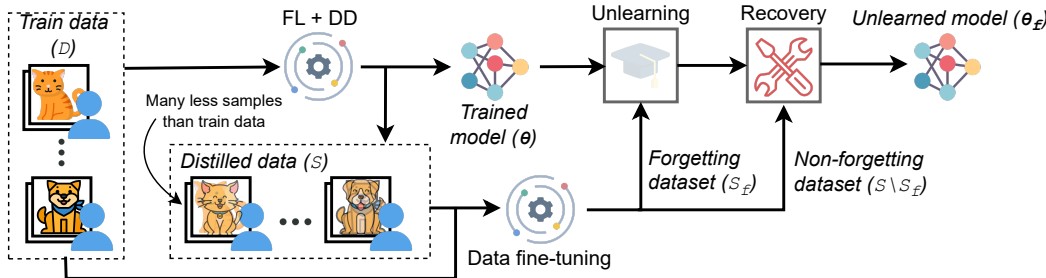

Figure 1: The workflow of QUICKDROP, our efficient unlearning method using distilled data.

This paper presents QUICKDROP, a novel FU approach that efficiently performs unlearning and recovery. QUICKDROP achieves this efficiency by harnessing dataset distillation (DD), a technique to condense a large training dataset into a compact synthetic dataset that is orders of magnitude smaller in volume (Wang et al., 2018). This synthetic dataset is generated such that it largely preserves the features of the original training dataset. QUICKDROP then utilizes this compact dataset during the unlearning and recovery phases. Figure 1 depicts the complete QUICKDROP workflow. Clients first engage in regular FL training with their train dataset to collaboratively produce a trained model. Each client also generates a compact synthetic dataset using DD. Upon receiving the initial unlearning request, the network executes unlearning rounds, during which each client performs SGA on their local distilled dataset. The network then executes recovery rounds, during which they also use the distilled data. The key to efficiency in QUICKDROP is the tiny volume of distilled data, resulting in efficient downstream unlearning compared to when using the original datasets of clients.

The distilled datasets can be generated independently from the FL training as the distilled dataset becomes necessary only after training finishes. More specifically, by leveraging a recent DD technique based on gradient matching (Zhao et al., 2021), QUICKDROP smartly re-uses the gradient updates produced during FL training for DD, thus significantly reducing the overhead of the DD process.

**Contributions.** This paper makes the following three key contributions:

1. We introduce QUICKDROP, a novel and efficient federated unlearning approach that leverages dataset distillation to unlearn specific knowledge from a trained model (Section 3).

2. To reduce the computational overhead of DD, we seamlessly integrate DD into FL by reusing the gradient updates generated during FL training for creating distilled datasets (Section 3.3).

3. We implement and open-source QUICKDROP, and evaluate its unlearning performance in terms of efficiency and accuracy on three standard datasets (Section 4). We find that QUICKDROP reduces the duration of class unlearning by **463.8×** compared to model retraining from scratch and **65.1×** compared to state-of-the-art FU approaches.

## 2 BACKGROUND AND RELATED WORK

**Federated Unlearning.** Cao & Yang (2015) first propose the concept of machine unlearning (MU) to eliminate the contribution of one specific training data sample from a well-trained model. Since then, many other algorithms for machine unlearning have been introduced (Du et al., 2019; Ginart et al., 2019; Guo et al., 2019; Golatkar et al., 2020a;b; Bourtoule et al., 2021). These works focus mainly on unlearning knowledge from linear classification models, *e.g.*, for logistic regression, but are unsuitable for more complex models, *e.g.*, deep neural networks. Some algorithms have other restrictions and can only be applied to specific model architectures or scenarios, *e.g.*, (Brophy & Lowd, 2021) only fits random forests model and (Nguyen et al., 2020) is only for Bayesian learning.

Federated Unlearning FU is a MU technique where knowledge is removed from a trained model in a distributed and collaborative manner. Generally, FU is more challenging than MU for the following two reasons. First, the model aggregation of FL intertwines the information of all clients, making it hard to identify and remove specific data samples. Secondly, client data is only available locally and cannot be moved to a central server, therefore mandating active participation by clients to perform unlearning. (Liu et al., 2021) propose FedEraser to achieve client-level unlearning, *i.e.*, eliminating the impact of data of one FL client from the FL global model. FedEraser reconstructs the global model by utilizing all historical update gradients of clients stored in the server during training. The

key idea here is to trade the additional storage cost for less computation cost (faster unlearning). (Wang et al., 2022a) mainly focus on class-level unlearning; they measure the class discrimination of channels in the model, *i.e.*, the relevance of different classes on the model channel) and then prune the most relevant channel of the target class to unlearn it. (Wu et al., 2022) proposed a more general federated unlearning framework by inverse gradient ascent, which achieves unlearning on class, client, and sample levels. However, this process remains inefficient, particularly when the volume of data is large or when multiple unlearning requests need to be executed.

**Dataset Distillation.** The goal of DD is to replace a large training dataset with a significantly smaller one that can substitute the original dataset during model training (Wang et al., 2018). DD can potentially speed up downstream tasks such as continual learning (Hadsell et al., 2020) and neural architecture search (Ren et al., 2021). QUICKDROP employs DD to speed up the unlearning process significantly. Early DD approaches are based on coreset selection that identifies a subset of influential samples during training. Sample selection, for example, can aim to maximize diversity (Aljundi et al., 2019), detect clusters of similar samples (Sener & Savarese, 2017), or identify individual samples that lead to high accuracy (Lapedriza et al., 2013). Another class of algorithms synthesizes a set of new samples from the original dataset. The approach described in (Zhao et al., 2021) is to match the gradients of a model trained on the original and synthetic data. Follow-up work has introduced distillation techniques based on trajectory gradient matching (Cazenavette et al., 2022), differential data augmentation functions (Zhao & Bilen, 2021), distribution matching (Zhao & Bilen, 2023) and feature alignment (Wang et al., 2022b). Other work utilizes DD for one-shot FL, which significantly reduces communication cost compared to multi-round FL (Song et al., 2023; Zhou et al., 2020).

## 3  DESIGN OF QUICKDROP

In this section, we first formally define our notation and problem setup. We then describe the unlearning algorithm in Section 3.1. Section 3.2 describes how we leverage DD to unlock efficient unlearning and presents the DD algorithm. Section 3.3 shows how QUICKDROP integrates DD with FL training before summarizing end-to-end QUICKDROP in Section 3.4.

**Problem Setup.**  We consider an FL system containing total $N$ clients (*e.g.*, mobile devices), where each client $i \in N$ holds a local training dataset $D_i$. The clients collaboratively train a global FL model $\theta$ following a standard FL algorithm (*e.g.*, FEDAVG (McMahan et al., 2017)). Once the global model $\theta$ is trained, the federated server may receive an unlearning request for the subset $D_f$. We refer to $D_f$ as the *forgetting dataset* that needs to be unlearned from the global model $\theta$. The characterization of $D_f$ defines the type of unlearning performed. For instance, when $D_f$ contains the data of an entire class, we perform class-level unlearning, whereas when $D_f$ contains a single client's data, we perform client-level unlearning. We define a FU algorithm $\mathcal{U}$ as $\theta_f = \mathcal{U}(\theta, D_f)$, where $\theta_f$ is the unlearned model. Unlearning aims to obtain a model $\theta_f$ that is equivalent in performance to a model trained only on the remaining dataset $D \backslash D_f$. In other words, the unlearning model $\theta_f$ should achieve good performance on $D \backslash D_f$ while showing poor performance on $D_f$.

### 3.1  THE UNLEARNING ALGORITHM

When the server receives an unlearning request for $D_f$, it initiates unlearning rounds that resemble traditional FL training rounds. However, each client now performs SGA instead of regular SGD on the portion of samples in $D_f$. In each round, the server aggregates the models received from clients. However, SGA training often introduces noise that affects the performance of remaining data (Wu et al., 2022). This noise necessitates subsequent recovery rounds during which clients engage in regular SGD training on the remaining data, *i.e.*, $D \backslash D_f$. We refer to $D \backslash D_f$ as the *recovery* set. As we experimentally show later, this recovery phase rapidly restores the accuracy of the remaining classes. The execution of an unlearning request, therefore, encompasses unlearning on $D_f$ and recovery on $D \backslash D_f$, thus updating the model with the entire dataset. Therefore, this process remains inefficient, in particular with high data volumes or in the presence of many unlearning requests.

### 3.2  DATASET DISTILLATION FOR EFFICIENT UNLEARNING

To significantly reduce the volume of data involved when executing an unlearning request, we utilize DD to condense all critical information from the original training dataset into a significantly smaller

synthetic or distilled dataset $S$. In our FL setting, each client $i \in N$ independently distills its local dataset $D_i$ into $S_i$ such that $|S_i| \ll |D_i|$. The unlearning algorithm $\mathcal{U}$ can thus be modified as $\boldsymbol{\theta}_f = \mathcal{U}(\boldsymbol{\theta}, S_f)$, where $S_f$ is the counterpart of the unlearning dataset $D_f$ in the distilled dataset. Since the distilled data is orders of magnitude smaller in volume, the unlearning task can be carried out very efficiently.

QUICKDROP adopts the recent algorithm of Zhao et al. (2021) to perform dataset distillation on each FL client. The primary reason for choosing this algorithm is the striking similarity of the algorithmic structure to standard FL algorithms. As one of the main contributions in this paper, we leverage this similarity to integrate the process of distilling data while training the FL model. This in-situ distillation avoids expensive computational overheads of distilling data separately.

**Dataset Distillation.** Before describing the algorithm of Zhao et al. (2021), we first formalize the task of DD. We adopt the DD formulation from Zhao et al. (2021). Suppose we are given a training dataset $\mathcal{D}$ with $|\mathcal{D}|$ pairs of a training images and class labels $\mathcal{D} = \{(\boldsymbol{x}_i, y_i)\}|_{i=1}^{|\mathcal{D}|}$ where $\boldsymbol{x} \in \mathcal{X} \subset \mathbb{R}^d$, $y \in \{0, \dots, C-1\}$, $\mathcal{X}$ is a d-dimensional input space and $C$ is the number of classes. The goal is to learn a differentiable function $\phi$ (*i.e.*, a deep neural network) with parameters $\boldsymbol{\theta}$ that correctly predicts labels of unseen images. One can learn the parameters of this function by minimizing an empirical loss term over the training set:

$$\boldsymbol{\theta}^{\mathcal{D}} = \underset{\boldsymbol{\theta}}{\arg\min} \, \mathcal{L}^{\mathcal{D}}(\boldsymbol{\theta}) \tag{1}$$

where $\mathcal{L}^{\mathcal{D}}(\boldsymbol{\theta}) = \frac{1}{|\mathcal{D}|} \sum_{(\boldsymbol{x},y)\in\mathcal{D}} \ell(\phi_{\boldsymbol{\theta}}(\boldsymbol{x}), y)$ , $\ell(\cdot, \cdot)$ is a task-specific loss (*i.e.*, cross-entropy) and $\boldsymbol{\theta}^{\mathcal{D}}$ is the minimizer of $\mathcal{L}^{\mathcal{D}}$. The generalization performance of the obtained model $\phi_{\boldsymbol{\theta}^{\mathcal{D}}}$ can be written as $\mathbb{E}_{\boldsymbol{x}\sim P_{\mathcal{D}}}[\ell(\phi_{\boldsymbol{\theta}^{\mathcal{D}}}(\boldsymbol{x}), y)]$ where $P_{\mathcal{D}}$ is the data distribution. The goal of DD is to generate a small set of condensed synthetic samples with their labels, $\mathcal{S} = \{(\boldsymbol{s}_i, y_i)\}|_{i=1}^{|\mathcal{S}|}$ where $\boldsymbol{s} \in \mathbb{R}^d$ and $y \in \mathcal{Y}$, $|\mathcal{S}| \ll |\mathcal{D}|$. Similar to eq. (1), one can train $\phi$ with these synthetic samples as follows:

$$\boldsymbol{\theta}^{\mathcal{S}} = \underset{\boldsymbol{\theta}}{\arg\min} \, \mathcal{L}^{\mathcal{S}}(\boldsymbol{\theta}) \tag{2}$$

where $\mathcal{L}^{\mathcal{S}}(\boldsymbol{\theta}) = \frac{1}{|\mathcal{S}|} \sum_{(\boldsymbol{s},y)\in\mathcal{S}} \ell(\phi_{\boldsymbol{\theta}}(\boldsymbol{s}), y)$ and $\boldsymbol{\theta}^{\mathcal{S}}$ is the minimizer of $\mathcal{L}^{\mathcal{S}}$. The goal of DD is to obtain $S$ such that the generalization performance of $\phi_{\boldsymbol{\theta}^{\mathcal{S}}}$ is as close as possible to $\phi_{\boldsymbol{\theta}^{\mathcal{D}}}$, *i.e.*, $\mathbb{E}_{\boldsymbol{x}\sim P_{\mathcal{D}}}[\ell(\phi_{\boldsymbol{\theta}^{\mathcal{D}}}(\boldsymbol{x}), y)] \simeq \mathbb{E}_{\boldsymbol{x}\sim P_{\mathcal{D}}}[\ell(\phi_{\boldsymbol{\theta}^{\mathcal{S}}}(\boldsymbol{x}), y)]$ over the real data distribution $P_{\mathcal{D}}$.

**DD with Gradient Matching.** The goal of obtaining comparable generalization performance by training on the condensed data can be formulated in multiple ways. Zhao et al. (2021) formulate the problem such that the model $\phi_{\boldsymbol{\theta}^{\mathcal{S}}}$ trained on $S$ achieves not only comparable generalization performance to $\phi_{\boldsymbol{\theta}^{\mathcal{D}}}$ but also converges to a similar solution in the parameter space (*i.e.*, $\boldsymbol{\theta}^{\mathcal{S}} \approx \boldsymbol{\theta}^{\mathcal{D}}$). This is achieved by matching gradients obtained over training and synthetic data over several time steps, thereby making $\boldsymbol{\theta}^{\mathcal{S}}$ follow a similar path to $\boldsymbol{\theta}^{\mathcal{D}}$ throughout the optimization. Precisely, $\mathcal{S}$ is obtained as minimizer of the following:

$$\underset{\mathcal{S}}{\min} \, \mathbb{E}_{\boldsymbol{\theta}_0 \sim P_{\boldsymbol{\theta}_0}} \left[ \sum_{t=0}^{T-1} d(\nabla_{\boldsymbol{\theta}} \mathcal{L}^{\mathcal{S}}(\boldsymbol{\theta}_t), \nabla_{\boldsymbol{\theta}} \mathcal{L}^{\mathcal{D}}(\boldsymbol{\theta}_t)) \right] \tag{3}$$

where $d(.;.)$ is a function that measures the distance between the gradients for $\mathcal{L}^{\mathcal{S}}$ and $\mathcal{L}^{\mathcal{D}}$ w.r.t $\boldsymbol{\theta}$. Since deep neural are sensitive to initialization, the optimization in Equation (3) aims to obtain an optimum set of synthetic images to generate samples that can work with a distribution of random initializations $P_{\boldsymbol{\theta}_0}$. At each time step, the synthetic data samples are updated by running a few local steps of an optimization algorithm `opt-alg` (*e.g.*, SGD):

$$\mathcal{S} \leftarrow \texttt{opt-alg}_{\mathcal{S}}(d(\nabla_{\boldsymbol{\theta}} \mathcal{L}^{\mathcal{S}}(\boldsymbol{\theta}_t), \nabla_{\boldsymbol{\theta}} \mathcal{L}^{\mathcal{D}}(\boldsymbol{\theta}_t)), \varsigma_{\mathcal{S}}, \eta_{\mathcal{S}}), \tag{4}$$

where $\varsigma_{\mathcal{S}}$ and $\eta_{\mathcal{S}}$ correspond to the number of steps and the learning rate. The model parameters are then updated through the loss on the updated synthetic data:

$$\boldsymbol{\theta}_{t+1} \leftarrow \boldsymbol{\theta}_t - \eta_{\boldsymbol{\theta}} \nabla \mathcal{L}^{\mathcal{S}}(\boldsymbol{\theta}_t) \tag{5}$$

where $\eta_{\boldsymbol{\theta}}$ is the learning rate for the update. The gradient matching continues for $T$ steps along the optimization path, progressively improving the quality of $S$. This process is repeated for several initializations $K$ drawn from $P_{\boldsymbol{\theta}_0}$. Algorithm 1, adapted from (Zhao et al., 2021), shows the complete distillation process.

**Algorithm 1:** Dataset condensation with gradient matching (Zhao et al., 2021).

**Input:** Randomly initialized synthetic samples $\mathcal{S}$, training dataset $\mathcal{D}$, outer-loop steps $K$, inner-loop steps $T$, steps for updating synthetic samples $\varsigma_{\mathcal{S}}$ in each inner-loop step, learning rate $\eta_{\boldsymbol{\theta}}$ and synthetic samples $\eta_{\mathcal{S}}$.

1 **for** $k = 0, \cdots, K-1$ **do**
2    —
3    Initialize $\boldsymbol{\theta}_0 \sim P_{\boldsymbol{\theta}_0}$
4    **for** $t = 0, \cdots, T-1$ **do**
5       Sample minibatch pairs $B^{\mathcal{D}} \sim \mathcal{D}$ and $B^{\mathcal{S}} \sim \mathcal{S}$
6       Compute $\nabla\mathcal{L}^{\mathcal{D}}(\boldsymbol{\theta}_t)$ on $B^{\mathcal{D}}$
7       Compute $\nabla\mathcal{L}^{\mathcal{S}}(\boldsymbol{\theta}_t)$ on $B^{\mathcal{S}}$
8       Update $\mathcal{S}$ using $d(\nabla\mathcal{L}^{\mathcal{D}}, \nabla\mathcal{L}^{\mathcal{S}})$ ▷ eq. (4)
9       ▷ Use the whole $\mathcal{S}$
10       $\boldsymbol{\theta}_{t+1} \leftarrow \boldsymbol{\theta}_t - \eta_{\boldsymbol{\theta}}\nabla\mathcal{L}^{\mathcal{S}}(\boldsymbol{\theta}_t)$
11    —

**Output:** $\mathcal{S}$

**Algorithm 2:** Integrating DD with FL training. Lines for FL are shown in purple.

**Input:** Randomly initialized synthetic samples $\{\mathcal{S}_i\}_{i=1}^N$, training datasets $\{D_i\}_{i=1}^N$, FL global rounds $K$, local update steps $T$, steps for updating synthetic samples $\varsigma_{\mathcal{S}}$ in an inner-loop step, learning rate $\eta_{\boldsymbol{\theta}}$ and synthetic samples $\eta_{\mathcal{S}}$.

1 **for** $k = 0, \cdots, K-1$ **do**
2    **for** *each client $i = 1, \cdots, N$ in parallel* **do**
3       Initialize $\boldsymbol{\theta}_0^i \leftarrow \boldsymbol{\theta}_k$
4       **for** $t = 0, \cdots, T-1$ **do**
5          Sample minibatch pairs $B^{\mathcal{D}_i} \sim \mathcal{D}_i$ and $B^{\mathcal{S}_i} \sim \mathcal{S}_i$
6          Compute $\nabla\mathcal{L}^{\mathcal{D}_i}(\boldsymbol{\theta}_t^i)$ on $B^{\mathcal{D}_i}$
7          Compute $\nabla\mathcal{L}^{\mathcal{S}_i}(\boldsymbol{\theta}_t^i)$ on $B^{\mathcal{S}_i}$
8          Update $\mathcal{S}_i$ using $d(\nabla\mathcal{L}^{\mathcal{D}_i}, \nabla\mathcal{L}^{\mathcal{S}_i})$ ▷ eq. (4)
9          ▷ Reuse the above gradient
10          $\boldsymbol{\theta}_{t+1}^i \leftarrow \boldsymbol{\theta}_t^i - \eta_{\boldsymbol{\theta}}\nabla\mathcal{L}^{\mathcal{D}_i}(\boldsymbol{\theta}_t^i)$
11    $\boldsymbol{\theta}_{k+1} \leftarrow \sum_{i=1}^N \frac{|\mathcal{D}_i|}{|\mathcal{D}|}\boldsymbol{\theta}_T^i$

**Output:** $\{\mathcal{S}_i\}_{i=1}^N, \boldsymbol{\theta}_K$

## 3.3 INTEGRATING DD WITH FL TRAINING BY RE-USING GRADIENTS

Producing good-quality synthetic data that can reliably be used as a substitute for the training data requires many local update steps. Thus, DD is computationally expensive. To reduce this overhead, we integrate the DD process with FL training such that clients reuse the gradient updates computed during FL training to perform DD. We achieve this by exploiting the algorithmic similarity of Zhao et al. (2021) (Algorithm 1) with standard FL algorithms. Algorithm 2 shows how QUICKDROP integrates DD with FL training, with the key differences from Algorithm 1 colored in red.

The outer loop of Algorithm 1 corresponds to global rounds in FL (line 1), while the inner loop corresponds to the local update steps in FL (line 4). The random initialization of model parameters in DD is substituted with initialization to the global model parameters received from the server in round $k$ (line 3). A main difference between Algorithm 1 and 2 is that Algorithm 1 distills using $K$ randomly initialized models, whereas Algorithm 2 only uses a single initialized model. This still works for unlearning as we do not prioritize generalization across different initializations in QUICKDROP. Our goal is instead to create a synthetic dataset with which can be utilized to unlearn knowledge from a model that was trained with a specific initialization. Clients generate gradients on mini-batches of their original training data to perform local updates (line 6). QUICKDROP effectively utilizes these gradients to update the synthetic data by matching the gradients computed on local synthetic data (lines 7-8). Finally, the model update step on synthetic data in DD is substituted by the local update, which the client performs for FL training (line 10). The federated server aggregates the received models before commencing the next round (line 11). Thus, QUICKDROP exploits gradients from FL training to efficiently generate distilled data for downstream unlearning. We note that the algorithm in Zhao et al. (2021) proposes a class-wise gradient matching which performs better than random batches. We also utilize it in QUICKDROP but omit from the pseudo code for presentation clarity.

**Fine-Tuning distilled data.** As the FL model training approaches convergence, the magnitude of gradients computed on both the training data and the synthetic data approaches zero. This prevents the synthetic data from being updated, resulting in slightly lower quality samples than when performing DD independently of FL. Therefore, to match the performance of independent DD, we let clients conduct additional fine-tuning steps to further improve the quality of their synthetic data. In this fine-tuning phase, QUICKDROP clients perform Algorithm 1 on their synthetic data.

## 3.4 END-TO-END WORKFLOW OF QUICKDROP

Finally, we summarize the end-to-end workflow of QUICKDROP, which is also depicted in Figure 1. Clients initially train a global model via FL while also conducting dataset distillation (DD)

to generate distilled datasets. For this, we use the integrated DD approach based on gradient matching described in Sections 3.2 and 3.3. The quality of the distilled dataset can be improved using fine-tuning. The distilled dataset is then employed for unlearning and recovery, also see Section 3.1.

**Mixing original data samples.** Clients also utilize the distilled dataset for the recovery phase. In our experiments, we observed that while using distilled data works well for unlearning, it slightly hurts the achieved performance in the recovery phase as the distilled samples are not a perfect representation of the original datasets. We found that even including a few original samples into the distilled datasets can nullify this performance drop. Thus, clients in QUICKDROP perform recovery on the merged dataset comprising distilled data and a few original samples.

## 4 EVALUATION

We evaluate QUICKDROP and compare its efficiency and performance with state-of-the-art FU baselines. We first describe the experimental setup and then show the unlearning performance of QUICK-DROP, the computational efficiency of our integrated FL and DD approach, the impact of additional fine-tuning steps, and finally the scalability of QUICKDROP with 100 clients.

### 4.1 EXPERIMENTAL SETUP

We evaluate the performance of QUICKDROP on three standard image classification datasets: MNIST (LeCun, 1998), CIFAR-10 (Krizhevsky et al., 2009), and SVHN (Netzer et al., 2011). For all datasets, we use a ConvNet as the deep neural network backbone (Gidaris & Komodakis, 2018). All experiments are conducted on a machine equipped with an i9-10900K CPU and an RTX 3090 GPU. All source code is available online. *Link omitted due to double-blind review.*

**Federated Learning.** To generate heterogeneous client local datasets, we adopt the Dirichlet distribution based approach from several previous works (Hsu et al., 2019; Zhu et al., 2021). The degree of non-IIDness is controlled by a parameter $\alpha \in [0, \infty)$, with lower values corresponding to more heterogeneity. In this section, we fix $\alpha = 0.1$ which is highly non-IID, and show experiments with IID distributions in Appendix B. We conduct all experiments in this section with ten clients, and we quantify the scalability of QUICKDROP with 100 clients on the SVHN dataset in Section 4.5. We use a full client participation in each round and train for 200 global rounds (*i.e.*, $K = 200$), sufficient to reach convergence. All other FL-related hyper-parameters follow McMahan et al. (2017).

**Dataset Distillation.** We initialize the synthetic samples $\{\mathcal{S}_i\}_{i=1}^{N}$ as randomly selected real training samples from the original local client dataset. We found this to be more effective in our setting than when initializing these samples from Gaussian noise. We use the same distance function $d$ for gradient matching as in Zhao et al. (2021). Our evaluation mainly focuses on class-level unlearning and shows extensive evaluations comparing QUICKDROP to state-of-the-art baselines. Throughout this section, we refer to the class(es) being unlearned as *target class(es)*. However, we note that QUICKDROP also supports client-level unlearning, which we experimentally show in Appendix C.

To guarantee that the distribution of the distilled dataset of each client reflects their original dataset distribution, we scale the number of distilled samples for each class for different clients by a factor $s$ (*i.e.*, distillated sample size per class $= \frac{\text{original data size per class}}{s}$). For any class with a distilled sample size of zero after scaling, we will round it up to 1 to ensure that this class has at least one distilled sample. We fix $s = 100$ for all experiments, which we found to yield a reasonable balance between efficiency and effectiveness. All other DD-related hyper-parameters follow Zhao et al. (2021).

**Baselines.** We compare the performance of QUICKDROP to the following three baselines:

1. **RETRAIN-OR**: This baseline retrains a model from scratch with FL using the original dataset minus the forgetting dataset.
2. **SGA-OR**: This baseline performs SGA using the original dataset (Wu et al., 2022). When the deletion request of a target class arrives, every involved client executes SGA on the data of the target class it owns to unlearn the contribution of the target class in the model.
3. **FU-MP**: This baseline uses model pruning by first measuring the class discrimination of channels in the model (*i.e.*, the relevance of different classes on the model channel) and then prunes the most relevant channel of the target class to unlearn it (Wang et al., 2022a).

All reported testing accuracies are the Top-1 accuracy achieved on the testing data of each dataset. We turn off fine-tuning for all experiments ($F = 0$), except for the experiments reported in Section 4.4. For all experiments with QUICKDROP, we perform a single round of unlearning and two rounds for recovery, which we found to be the minimum for the model to sufficiently unlearn particular knowledge and restore the performance of the remaining classes. Doing more unlearning rounds would introduce noise into the model and lower the accuracy of the remaining classes, making it more difficult to recover them later. We run each experiment five times and average all results. Additional notes on the experimental setup and parameters can be found in Appendix A.

## 4.2 PERFORMANCE EVALUATION ON A SINGLE UNLEARNING REQUEST

**Unlearning a Single Class.** We first quantify the change in testing accuracy of target and non-target classes after the unlearning and recovery stages, using the CIFAR-10 dataset and 10 clients. The network collaboratively unlearns from the model the knowledge corresponding to class 9 (digit "9") by performing one round of unlearning and two rounds of recovery. Figure 2 shows the testing accuracy for each class over time in different colors when unlearning with QUICK-DROP. When QUICKDROP starts the unlearning stage, we observe a rapid accuracy drop on the target class while the accuracy of non-target classes decreases as well. This is because SGA introduces some noise that affects non-target classes, even though the model parameters changed by SGA are mainly for unlearning

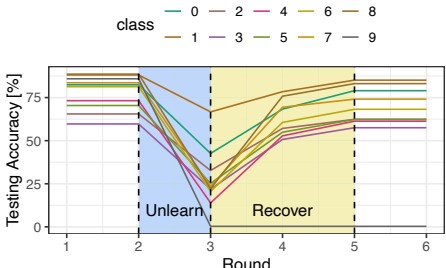

Figure 2: The testing accuracy for all classes with CIFAR-10 when unlearning class 9 and recovering the accuracy of the other classes with QUICKDROP.

the knowledge of the target class. The recovery stage that starts at round 3, therefore, restores the accuracy of non-target classes by training the global model on all distilled data of the remaining classes using stochastic gradient descent (SGD). Figure 2 shows that the accuracies of non-target classes after two recovery rounds are almost restored to their original values.

**Testing Accuracy.** Next, we compare the testing accuracy of QUICKDROP with our baselines on CIFAR-10 when unlearning a single class. Table 1 shows the testing accuracy on the F-Set and R-Set for different FU approaches and after each stage (unlearning and recovery). The *F-Set* is the set of samples being unlearned, and the *R-Set* is comprised of all other samples. Ideally, any FU approach should achieve near-zero testing accuracy on the F-Set after unlearning. We remark that there is no recovery stage for RETRAIN-OR. Table 1 shows that, after the unlearning stage, all approaches effectively eliminate the knowledge of the target class in the model as the F-Set testing accuracy is near zero (0.85% for QUICKDROP). After recovery, SGA-OR and FU-MP restore the accuracy of the R-Set close to the values achieved with RETRAIN-OR. The accuracy of QUICKDROP after recovery, 71.98%, is lower than that of the baselines. This is because the synthetic samples generated do not represent the original dataset perfectly. However, additional fine-tuning of distilled datasets can close this gap, at the cost of additional computation (see Section 4.4). Nonetheless, we conclude that QUICKDROP effectively unlearns data samples with minimal impact on the remaining samples.

**Computation Efficiency.** Table 1 also shows the computational cost for unlearning and recovery in terms of rounds, time required, and data samples involved in executing these rounds. We observe significant differences in computation cost between different FU approaches. Since DD reduces the number of samples for each client used during unlearning, the unlearning stage (one round) in

| Stage | Unlearning | | | | | Recovery | | | | |
|---|---|---|---|---|---|---|---|---|---|---|
| | Testing Accuracy | | Computation Cost | | | Testing Accuracy | | Computation Cost | | |
| **FU approach** | F-Set | R-Set | Round | Time (s) | Data Size | F-Set | R-Set | Round | Time (s) | Data Size |
| QUICKDROP | 0.82% | 37.68% | 1 | 5.03 | 100 | 0.85% | 70.48% | 2 | 10.58 | 900 |
| RETRAIN-OR | 0.81% | 74.95% | 30 | 7239.58 | 45 000 | — | — | — | — | — |
| SGA-OR | 0.75% | 48.69% | 2 | 495.17 | 5000 | 1.03% | 74.83% | 2 | 551.33 | 45 000 |
| FU-MP | 0.12% | 11.58% | 1 | 61.36 | 50 000 | 0.09% | 73.96% | 4 | 953.62 | 45 000 |

Table 1: The test accuracy and computation cost of QUICKDROP and FU baselines on the CIFAR-10 dataset, with non-IID data distributions ($\alpha = 0.1$) and in a 10-client network.

QUICKDROP only takes $5.03$ s, and $10.58$ s in the recovery stage; both stages are completed in just $15.61$ s. This efficiency is because a round of unlearning and recovery with QUICKDROP only involves 100 and 900 data samples, respectively. Although SGA-OR only needs two rounds to unlearn a target class adequately, it takes $247.58$ s to complete a round as completing this round updates the model with all clients' original data (5000 data samples in the unlearning stage and $45\,000$ samples in the recovery stage). While RETRAIN-OR is the simplest method with high testing accuracies after unlearning, its computational time renders this approach infeasible in many scenarios, which is $1447\times$ higher than QUICKDROP and $14\times$ higher than SGA-OR. We note that FU-MP employs a different technique, model pruning, than the other approaches in its unlearning stage, while the recovery stage is the same as others. Since model pruning only depends on information obtained from inference, which can be done relatively quickly, the time to complete a single round ($61.36$ s) is relatively small compared to SGA-OR ($247.58$ s), but its gap with QUICKDROP ($5.03$ s) is still significant. From Table 1 we conclude that QUICKDROP achieves quick unlearning and recovery using only a few data samples and with little computational overhead.

QUICKDROP is not only suitable for efficient unlearning but can also be applied for efficient relearning when the deleted knowledge is needed again. We use the distilled datasets to relearn particular samples and present experiments showing the performance of unlearning in Appendix B.

**Membership Inference Attack.** We conduct a membership inference attack (MIA) on the unlearned model to further assess the effectiveness of unlearning with QUICKDROP, which follows related work on MU Chen et al. (2021). The MIA aims to determine whether a particular sample is included in the model's knowledge. We implement the MIA according to the settings in Golatkar et al. (2021) and measure how often the attack model will classify a data

| Stage | Unlearning | | Recovery | |
|---|---|---|---|---|
| **FU approach** | F-Set | R-Set | F-Set | R-Set |
| QUICKDROP | 0.93% | 53.39% | 0.81% | 71.62% |
| RETRAIN-OR | 0.37% | 77.25% | — | — |
| SGA-OR | 0.68% | 65.71% | 0.72% | 74.21% |
| FU-MP | 0.79% | 37.68% | 0.94% | 72.56% |

Table 2: The membership inference attack (MIA) accuracy of all baselines in different stages.

sample from deleted data as a training sample of the unlearning model. Table 2 shows these MIA accuracies of different methods and stages. The performance of RETRAIN-OR can be considered optimal since the produced model has never seen the unlearned samples. We find that for all approaches, the MIA accuracy on the F-Set after the unlearning stage is below 1%. The MIA accuracy on the R-Set for evaluated FU approaches show similar trends as the values observed in Table 1, *i.e.*, QUICKDROP shows a slightly lower MIA accuracy on the R-Set compared to the other baselines.

### 4.3 THE PERFORMANCE OF QUICKDROP WITH SEQUENTIAL UNLEARNING REQUESTS

In real-world settings, clients may continually launch multiple, sequential unlearning requests. Therefore, we go beyond existing work on FU and evaluate the performance of QUICKDROP with sequential unlearning requests. Figure 3 shows the accuracies when sequentially unlearning all ten CIFAR-10 classes in random order. We observe that the accuracy of a target class drops to near zero after the unlearning phase for that target class ends, illustrating the effectiveness of QUICK-DROP in handling sequential unlearning requests. The accuracies of non-target classes also drop after the unlearning stage, which is due to the noise introduced by SGA. In the subsequent recovery stages, QUICKDROP rapidly recovers the accuracies of remaining classes by training on the distilled

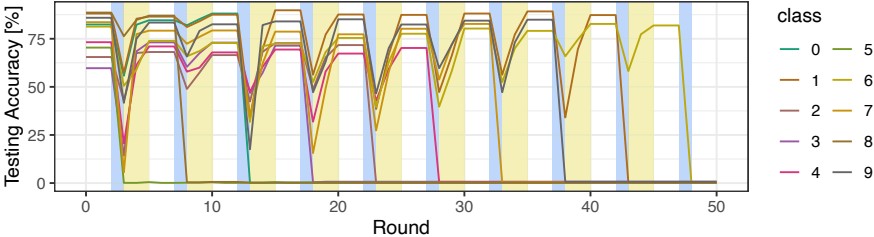

Figure 3: The testing accuracy for each class on sequential unlearning requests with CIFAR-10 and $\alpha = 0.1$. We unlearn a random class every five rounds and highlight the learning and recovery stages in blue and orange, respectively. The class unlearning order is $[5, 8, 0, 3, 2, 4, 7, 9, 1, 6]$.

datasets while leaving the low accuracy of the unlearned classes unaffected. Therefore, Figure 3 shows the capability of QUICKDROP in executing multiple unlearning requests. In particular situations, the network can process multiple unlearning requests in parallel. We discuss this optimization in Appendix D.

### 4.4 DATASET DISTILLATION AND ADDITIONAL FINE-TUNING

As discussed in Section 3.2, integrating DD into FL lowers the quality of the distilled dataset compared to when conducting FL and DD separately. To offset this, QUICKDROP allows clients to perform additional fine-tuning steps ($F$) by executing Algorithm 1. Figure 4 (left) shows the testing accuracy of QUICKDROP on the R-Set after the recovery stage when doing more fine-tuning (*i.e.*, increasing $F$ from 0 to 200). We also show the testing accuracy reached when separately performing DD and FL (74.78%), which we consider as an optimal baseline with a dashed horizontal line. We observe an increase in accuracy as $F$ in-

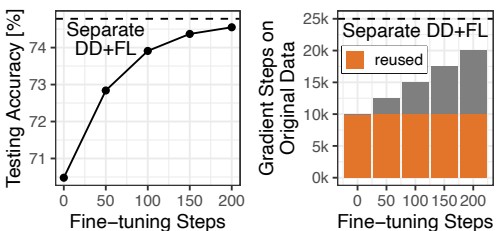

Figure 4: The testing accuracy on the R-Set after recovery (left) and gradient steps on original data (right) when doing additional fine-tuning.

creases: from 70.48% with $F = 0$ to 74.55% with $F = 200$. More fine-tuning, however, comes at the cost of additional computation. Figure 4 (right) shows the number of gradient steps performed on the original dataset as $F$ increases. This figure marks the portion of gradients we re-use for FL and DD in orange, which indicates the savings in computation by integrating DD and FL. When performing FL and DD separately, we require 25 000 gradient steps on the original dataset (indicated by a dashed line). However, when integrating FL and DD, we only need 10 000 gradient steps on the original dataset with $F = 0$, and we can re-use all these gradients for DD. With $F = 200$, this number increases to 20 000.

### 4.5 PERFORMANCE AND EFFICIENCY OF QUICKDROP IN LARGER NETWORKS

Finally, we analyze the unlearning performance of QUICKDROP and other baselines in a 100-client network and with the SVHN dataset. SVHN is a large dataset containing more than 600 000 samples and 10 classes. In each round, the server selects a random 10% subset of clients to update the model. Table 3 shows for each approach and stage the testing accuracy on the F-Set and R-Set when unlearning class 9. Even with 100 clients, QUICKDROP effec-

| Stage | Unlearning | | Recovery | |
|---|---|---|---|---|
| **Baseline** | F-Set | R-Set | F-Set | R-Set |
| QUICKDROP | 0.72% | 58.57% | 0.81% | 84.96% |
| RETRAIN-OR | 0.34% | 88.39% | — | — |
| SGA-OR | 0.53% | 52.48% | 0.66% | 86.47% |
| FU-MP | 0.58% | 39.46% | 0.73% | 85.63% |

Table 3: Testing accuracy of different FU approaches with 100 clients and the SVHN dataset.

tively unlearns class knowledge, and the accuracy after the unlearning stage on the F-set is just 0.72%. Compared to other baselines, QUICKDROP shows good testing accuracy after the recovery stage on the R-Set, even with more clients and samples in the training dataset. Additional fine-tuning steps can further reduce this gap. We also observe that QUICKDROP executes a complete unlearning request $475.2\times$ faster than RETRAIN-OR, highlighting the superior advantage in computation efficiency stemming from dataset distillation.

## 5 CONCLUSION

In this paper, we introduced QUICKDROP, a novel and efficient federated unlearning method that incorporates dataset distillation to address the challenges of erasing data from a trained ML model. QUICKDROP has clients produce and use a compact, distilled dataset that drastically reduces computational overhead during the unlearning and recovery phases. QUICKDROP elegantly combines the gradient matching DD and FL processes, allowing for gradient reuse and thereby further reducing computational overhead. Empirical evaluations on three standard datasets confirmed the effectiveness and efficiency of QUICKDROP, demonstrating a remarkable acceleration in the unlearning process compared to existing federated unlearning approaches.

REPRODUCIBILITY STATEMENT

We have undertaken several steps to ensure the integrity, reproducibility and replicability of QUICK-DROP. We have provided an extensive description of the proposed QUICKDROP in the main text, specifying its workflow in Figure 1, its theoretical formulation in 3 and the integration with Zhao et al. (2021) in Algorithm 2. To facilitate the reproducibility of the experimental results, the complete source code used for the evaluation of QUICKDROP will be made publicly available and a link will be added in the final paper version. We have used publicly available ML models and datasets. The details provided in Section 4, as well as the information provided in Section A, should be sufficient to reproduce our results. We believe that these efforts will aid researchers in understanding, replicating, and building upon QUICKDROP.

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
