## A    ADDITIONAL NOTES ON EXPERIMENTAL SETUP

We have tested QUICKDROP and baselines with a commonly used deep neural network architecture, *i.e.*, ConvNet (Gidaris & Komodakis, 2018). Its modular architecture contains $D$ duplicate blocks, and each block has a convolutional layer with $W$ ($3 \times 3$) filters, a normalization layer $N$, an activation layer $A$, and a pooling layer $P$, denoted as $[W, N, A, P] \times D$. The default ConvNet (unless specified otherwise) includes 3 blocks, each with 128 filters, followed by InstanceNorm, ReLU and AvgPooling modules. The final block is followed by a linear classifier.

After the integration of DD and FL training, we have the following hyper-parameters – the number of FL global rounds $K$, the number of local update steps $T$, the number of optimization step $\varsigma_{\mathcal{S}}$ for synthetic sample updating, and the learning rate $\eta_{\mathcal{S}}$. In all experiments, we set $K = 200$, $T = 50$, $\varsigma_{\mathcal{S}} = 1$, $\eta_{\mathcal{S}} = 0.1$. The number of FL global rounds is set to $K = 200$ since our trials indicated that the model will converge before that. Other hyperparameters follow previous work (Zhao et al., 2021). In mini-batch sampling for DD with gradient matching, we randomly sample 256 real images of a class as a mini-batch to calculate the gradients. We employ Stochastic Gradient Descent (SGD) as the optimizer.

For the evaluation of independent DD and FL, the definition of hyperparameters is slightly different — we have the number of outer-loop steps $K$, the number of inner-loop steps $T$, the number of optimization steps $\varsigma_{\mathcal{S}}$ for synthetic sample updating, and the learning rate $\eta_{\mathcal{S}}$. We set $K = 500$, $T = 50$, $\varsigma_{\mathcal{S}} = 1$, $\eta_{\mathcal{S}} = 0.1$ following the same settings in previous work (Zhao et al., 2021).

## B    ADDITIONAL RESULTS WITH DIFFERENT DATASETS, NETWORK SIZES, AND RELEARNING

In this section, we present additional accuracy results on a single unlearning request with different datasets (MNIST and CIFAR-10) and network sizes (10 and 20 clients). Since we already included the results on CIFAR-10 with 10 clients in Table 1, we include in this section the remaining combinations in Table 4 (CIFAR-10 with 20 clients), Table 5 (MNIST with 10 clients), and Table 6 (MNIST with 20 clients), respectively. All these experiments follow the same setup as the results shown in Table 1, *e.g.*, we use a non-IID data distribution ($\alpha = 0.1$). We also attach the additional results of the Relearning stage in each Table to show the effectiveness of different methods in relearning the eliminated knowledge again. The approach used in the relearning stage is the same for different baselines, we adopt the traditional SGD-based model training to update the "unlearning model" over the rejoined data. Note that our QUICKDROP still uses the distilled data in the relearning stage while other baselines use the original data, thus QUICKDROP can still keep its superiority in computation efficiency.

| Stage | Unlearning | | Recovery | | Relearning | |
|---|---|---|---|---|---|---|
| **FU approach** | F-Set | R-Set | F-Set | R-Set | F-Set | R-Set |
| QUICKDROP | 0.53% | 31.52% | 0.69% | 65.78% | 74.39% | 66.21% |
| RETRAIN-OR | 0.68% | 71.48% | — | — | 78.65% | 71.83% |
| SGA-OR | 0.49% | 25.37% | 0.71% | 70.04% | 75.83% | 69.75% |
| FU-MP | 0.43% | 20.43% | 0.59% | 69.82% | — | — |

Table 4: The testing accuracies for **CIFAR-10** and **20 clients** on the F-Set and R-Set, after the unlearning, recovery and relearning stages.

For all reported combinations of dataset and network sizes, we observe that all methods effectively eliminate the knowledge of a target class from the model as the testing accuracy on the F-Set is near-zero after the unlearning stage. Then, after the recovery stage, QUICKDROP, SGA-OR, and FU-MP all restore the accuracy of the R-Set close to the value on RETRAIN-OR. Consistent with our observations from Table 1, the accuracy on the R-Set by QUICKDROP after the recovery stage is slightly below the baselines. This is because the distilled data is not a perfect representation of the original training data. This accuracy gap can be reduced by additional fine-tuning of the distilled dataset, at the expense of computation overhead.

| Stage | Unlearning | | Recovery | | Relearning | |
|---|---|---|---|---|---|---|
| **FU approach** | F-Set | R-Set | F-Set | R-Set | F-Set | R-Set |
| QUICKDROP | 0.53% | 68.24% | 0.61% | 94.16% | 96.39% | 94.73% |
| RETRAIN-OR | 0.47% | 96.28% | — | — | 97.36% | 96.75% |
| SGA-OR | 0.41% | 71.38% | 047% | 96.26% | 97.05% | 95.82% |
| FU-MP | 0.28% | 66.45% | 0.44% | 95.37% | — | — |

Table 5: The testing accuracies for **MNIST** and **10 clients** on the F-Set and R-Set, after the unlearning, recovery and relearning stages.

| Stage | Unlearning | | Recovery | | Relearning | |
|---|---|---|---|---|---|---|
| **FU approach** | F-Set | R-Set | F-Set | R-Set | F-Set | R-Set |
| QUICKDROP | 0.33% | 64.58% | 0.44% | 94.26% | 96.37% | 94.58% |
| RETRAIN-OR | 0.47% | 95.63% | — | — | 96.82% | 95.74% |
| SGA-OR | 0.38% | 73.57% | 0.51% | 95.03% | 96.28% | 95.18% |
| FU-MP | 0.26% | 58.36% | 0.31% | 94.83% | — | — |

Table 6: The testing accuracies for **MNIST** and **20 clients** on the F-Set and R-Set, after the unlearning, recovery and relearning stages.

Table 4-6 also report the testing accuracy on the F-Set and R-Set after relearning. Ideally, we want these accuracies to be high since we attempt to restore the model the state before unlearning. Table 4-6 show that all evaluated FU approaches successfully relearn the previously eliminated knowledge again, while our QUICKDROP can still keep its superiority in computation efficiency since the relearning stage uses the compact, distilled dataset ($66.7\times$ faster than RETRAIN-OR and $47.29\times$ than SGA-OR). We are unable to relearning using FU-MP. This is because the unlearning method of FU-MP is based on model pruning, which irreversibly destroys the model structure during the unlearning stage. In particular, all channels related to the target class are pruned, and it is impossible to recover the knowledge of that particular class with such a damaged model.

## C  CLIENT-LEVEL UNLEARNING EVALUATION

Our evaluation in Section 4 establishes the performance of QUICKDROP and baselines when performing class-level unlearning. We now evaluate the effectiveness of QUICKDROP when performing client-level unlearning. The goal of client-level unlearning is to erase the data samples of a specific *target client* from the trained model. Being able to quickly perform client-level unlearning is essential to adhere to privacy regulations such as the right to be forgotten European Union (2018). We illustrate the performance of our QUICKDROP on client-level unlearning by comparing it with other baselines. FU-MP is unable to perform client-level unlearning as this approach is specifically designed for class-level unlearning. We conduct experiments on the CIFAR-10 dataset using two different data distributions: Non-IID ($\alpha = 0.1$) and IID (uniform distribution). The target unlearning client is selected randomly from all available clients and we reset the random seed to change the data distribution of clients in each run of the experiments.

Table 7 shows the results on client-level unlearning with Non-IID distribution (using $\alpha = 0.1$). This table shows that for all evaluated FU approaches, the testing accuracy on the F-Set after unlearning is not near zero (8.37% for QUICKDROP), unlike when doing class-level unlearning (see Table 1). The reason for this is that even though we unlearned the data samples of a particular client, some features associated with the classes that a particular target client holds might still be embedded in the model's knowledge. Because of this, it happens that these forgotten samples are correctly classified, even after unlearning. Furthermore, a target client $t$ may have the majority of data for a particular class $c$, while it only holds small amounts of data for other classes. Therefore, the model performance after the recovery stage now critically depends on the individual data distribution of clients as unlearning the data of client $t$ may significantly hurt the model performance on class $c$. Conversely, the knowledge on the classes of which the target client holds a small amount of data

| Distribution | Non-IID ($\alpha = 0.1$) | | | | | |
|---|---|---|---|---|---|---|
| **Stage** | Unlearning | | Recovery | | Relearning | |
| **FU approach** | F-Set | R-Set | F-Set | R-Set | F-Set | R-Set |
| QUICKDROP | 8.37% | 32.59% | 11.57% | 70.89% | 71.88% | 71.61% |
| RETRAIN-OR | 10.48% | 73.69% | — | — | 72.94% | 74.51% |
| SGA-OR | 6.72% | 21.49% | 9.58% | 72.63% | 72.49% | 73.53% |

Table 7: The testing accuracy of QUICKDROP and other baselines for client-level unlearning on **CIFAR-10** with a **Non-IID** ($\alpha = 0.1$) data distribution.

will not be completely eliminated from the model. Table 7 also shows that after the recovery stage, the testing accuracy on the R-Set (70.89%) is a bit lower than, but close to the performance of RETRAIN-OR on the R-Set (73.69%). These results are consistent with the accuracies obtained for class-level unlearning.

| Distribution | IID | | | | | |
|---|---|---|---|---|---|---|
| **Stage** | Unlearning | | Recovery | | Relearning | |
| **FU approach** | F-Set | R-Set | F-Set | R-Set | F-Set | R-Set |
| QUICKDROP | 30.53% | 34.29% | 68.59% | 68.48% | 71.57% | 70.69% |
| RETRAIN-OR | 70.81% | 71.64% | — | — | 71.40% | 74.08% |
| SGA-OR | 28.76% | 37.85% | 69.32% | 70.25% | 72.31% | 73.94% |

Table 8: The testing accuracy of QUICKDROP and other baselines for client-level unlearning on **CIFAR-10** with an **IID** data distribution.

Table 8, shows the results for client-level unlearning with IID data distributions. Similar to the results in Table 7, we observe relatively high accuracies on the F-Set after the unlearning stage. When comparing the accuracies of RETRAIN-OR on the F-Set (70.81%) and R-Set (71.64%), we find that unlearning the data samples of the target client has minimal impact on the overall model performance. This is because with an IID distribution, each client holds the same number of data samples for all classes. Therefore, when we unlearn the target client, much of its contributed knowledge is still represented by the remaining data (R-Set) in the system and the departure of the target client will barely impact the model performance.

**Sample-level Unlearning.** So far, we have shown the effectiveness and efficiency of QUICKDROP when performing class-level and client-level unlearning. These two levels of unlearning already cover many applications of machine unlearning. One might want to perform sample-level unlearning, where the goal is to unlearn a subset of data samples of a particular client. This is difficult to achieve with QUICKDROP since each client creates a distilled dataset that contains the knowledge of individual training samples in a compressed format. Even though the SGA algorithm can be performed with a subset of a client's samples, the recovery phase cannot be performed with the distilled dataset as this dataset again contains the knowledge of the samples being unlearned. Therefore, we consider this challenge beyond the scope of our work. However, we remark that QUICKDROP can be used to unlearn *all* samples of a particular class that a client holds since distilled datasets on the granularity of a class.

## D    EXECUTING MULTIPLE UNLEARNING REQUESTS IN PARALLEL

In Section 4.3, we have demonstrated how QUICKDROP is able to execute subsequent, multiple unlearning requests for different classes. While we assume in this experiment that unlearning requests are processed one-by-one, batching multiple unlearning requests could save time and compute resources. QUICKDROP supports the processing of multiple unlearning requests at the same time by having clients execute SGA using the distilled data representing the samples being unlearned, and then execute the recovery stage with the distilled data representing the remaining data. This enables

the network to unlearn multiple classes, or the data of multiple clients using a single unlearning and recovery stage.