# OpenReview forum: "QuickDrop: Efficient Federated Unlearning by Integrated Dataset Distillation"
_ICLR.cc/2024/Conference — Submitted to ICLR 2024_

### Official Review · Reviewer_BiDM · 2023-10-14

**Soundness:** 3 good
**Presentation:** 4 excellent
**Contribution:** 3 good
**Rating:** 6
**Confidence:** 4

**Summary:**

This work studies how to extract a much smaller dataset for federated unlearning (FU) using dataset distillation (DD), which speeds up the FU time. The method is incremental to (Zhao et al., 2021), adapting it to the federated setting with many clients. This method reuses the gradient updates produced during FL training for DD, so the overhead of creating distilled datasets is very small. However, the result quality is not as good as the more expensive baselines, but finetuning steps on the original dataset can mitigate this issue to achieve comparable accuracy.

**Strengths:**

The approach is effective especially in speeding up the unlearning request processing, and the idea makes sense and is intuitive.
The experiments are comprehensive.

**Weaknesses:**

The idea is very incremental to (Zhao et al., 2021). In particular, the main technical sections are Sec 3.2 and 3.3, but Sec 3.2 is basically reviewing (Zhao et al., 2021). Sec 3.3 is short and describes the small changes of (Zhao et al., 2021) in Algorithm 1 to achieve Algorithm 2, where the changes are very standard migration to the FedAvg setting.

The accuracy of QUICKDROP is lower than that of the baselines, and a solution of finetuning is proposed in Figure 4 to be effective. However, finetuning is claimed to be done on the original dataset. Does that mean you still need all clients to collaborate to complete the finetuning?

Finally, while the method is intuitive, it is very simple without much theory.

**Questions:**

Finetuning is claimed to be done on the original dataset. Does that mean you still need all clients to collaborate to complete the finetuning?

---

> ### Author Response · Authors · 2023-11-15
> **About the incremental  contribution on  (Zhao et al., 2021).**
>
> We understand the perceived limitation of our technical contribution. While it is true that our algorithm builds upon the gradient matching algorithm introduced by Zhao et al. (2021), it is not immediately evident if DD performs optimally in the context of FU. For instance, Zhao et al. (2021) train over several model initializations deemed necessary for good distillation, whereas only one model initialization is available for distillation in QuickDrop.
>
> Our contributions extend beyond the combination of existing methods in extensively demonstrating the viability, challenges, and modifications necessary to achieve efficient unlearning. Furthermore, our work emphasizes minimizing the overhead associated with DD generation, addressing a critical aspect often overlooked in previous studies.

---

> ### Author Response · Authors · 2023-11-15
> **Fine-tuning issues**
>
> We want to clarify that each client performs fine-tuning independently using its local dataset. That is, each client first generates its own distilled dataset during model training with FL and later locally fine-tunes after the FL training is over. Therefore, no collaboration is needed during the fine-tuning phase. We will clarify this in the final version of the paper.

---

> > ### Comment · Reviewer_BiDM · 2023-11-22
> > **Thanks for the response**
> >
> > Thanks for the response!

---

### Official Review · Reviewer_ug1m · 2023-10-28

**Soundness:** 2 fair
**Presentation:** 3 good
**Contribution:** 1 poor
**Rating:** 3
**Confidence:** 4

**Summary:**

This paper addresses the problem of federated unlearning and proposes an efficient method based on dataset distillation. It is used to accelerate both the training and unlearning stages The authors conduct unlearning by stochastic gradient ascent. The proposed method is evaluated experiments on three datasets.

**Strengths:**

1.	Federated unlearning is an interesting research topic. The authors also focus on a general framework of sample-wise unlearning, which is also applicable to class-wise and client-wise.
2.	The paper is clearly written and well organized.

**Weaknesses:**

1.	The technical contribution is incremental. The proposed method uses a gradient ascent + fine-tuning framework, which has been introduced in [1, 2]. The main contribute of this paper is the use of data distillation (a direct application of existing algorithm), which is similar to the idea of representative data selection in [3]. Thus, this paper seems to be a combination of existing methods.
2.	The proposed method is applicable to sample-wise unlearning, but the experiments are conducted on class-wise and client-wise unlearning. I was expected to see the evaluation of sample-wise unlearning, since it has broader applicability.
3.	Lack of baseline methods. I suggest that the authors consider including some representative ones or at least one method from more different approaches. Additionally, PU-MP is a CNN-specific method. It may not be appropriate to limit the target model to CNN and compare it with the proposed method, which is more general. I list some of the approaches for your consideration:
i.	Efficient retraining: [4, 5]
ii.	Influence function: [6]
iii.	Gradient ascent + regularization: [7, 8] ([8] is compared in the paper)
iv.	Roll back gradient + knowledge distillation: [1]
v.	Scaling: [9]

[1] 2022, Federated Unlearning with Knowledge Distillation
[2] AAAI’21, Amnesiac Machine Learning
[3] IJCAI’22, ARCANE：An Efficient Architecture for Exact Machine Unlearning
[4] IWQOS’21, FedEraser: Enabling Efficient Client-Level Data Removal from Federated Learning Models
[5] WSDM’23, Federated Unlearning for On-Device Recommendation
[6] INFOCOM’22, The Right to be Forgotten in Federated Learning: An Efficient Realization with Rapid Retraining
[7] ICML’22, Federated Unlearning: How to Efficiently Erase a Client in FL
[8] IEEE Network, Federated Unlearning: Guarantee the Right of Clients to Forget
[9] 2023, VERIFI: Towards Verifiable Federated Unlearning

**Questions:**

Please refer to Weaknesses.

---

> ### Author Response · Authors · 2023-11-15
> **Technical contribution**
>
> We appreciate the reviewer's examination of our proposed method and references to related work. While some aspects of our approach are built upon previous work, we argue that our work extends beyond a mere combination of existing methods. Although SGA and DD techniques are well-known, it is not immediately evident if their combination can perform well for FU. We conducted thorough empirical assessments on three diverse datasets to understand and demonstrate the effectiveness and adaptability of our approach across different unlearning scenarios, e.g., unlearning on class and client levels. Furthermore, our work emphasizes minimizing the overhead associated with DD generation, addressing a critical aspect often overlooked in previous studies.
>
> **Essential difference between DD and representative data selection**:
>
> Besides, we have carefully studied the related work indicated by the reviewer ([3]) about representative data selection and find there is a key difference between our approach and the representative data selection.
> DD in QuickDrop can absorb all the crucial information from the whole original dataset by condensation, while the approach in [3] can only choose part of the dataset and lose extensive information in those unselected data. The unlearning performance with representative data selection therefore has a worse accuracy degradation compared with QuickDrop. For example, Figure 6 in [3] reveals that if 25% data are selected from MNIST as representative dataset, the accuracy on the representative dataset is merely 60%, compared with 95% when using the full original dataset. However, in QuickDrop with scale=100 (i.e., the size of the distilled dataset is only 1% compared with the original dataset), the achieved test accuracy with the distilled dataset is 91%.

---

> ### Author Response · Authors · 2023-11-15
> **Sample-wise unlearning**
>
> We agree that sample-wise unlearning presents a more general scenario compared to class- and client-wise unlearning. While it would be valuable to tackle sample-wise unlearning, our framework in its current form requires additional adaptations to achieve this, and we should have been more clear about this limitation. We currently generate class-wise distilled samples that can be reused for downstream class-wise or client-wise unlearning. To achieve sample-wise unlearning, we would need to understand how to generate targeted distilled data for specific forgetting samples, and we left this out of this submission due to the lack of space and to keep the discussion more focused. We look forward to extending our work with sample-wise unlearning.

---

> ### Author Response · Authors · 2023-11-15
> **Lack of baseline methods.**
>
> Because of the time limitation, we can only add one more baseline for comparison during this rebuttal period, which is “IJCAI’22, ARCANE：An Efficient Architecture for Exact Machine Unlearning”. We will report these experiment results soon.

---

### Official Review · Reviewer_AkUq · 2023-10-29

**Soundness:** 2 fair
**Presentation:** 2 fair
**Contribution:** 2 fair
**Rating:** 3
**Confidence:** 3

**Summary:**

The paper proposes QuickDrop, a new federated unlearning (FU) approach that uses dataset distillation to efficiently remove specific data from a collaboratively trained machine learning model. QuickDrop has each client create a highly condensed "distilled" dataset that preserves the key features of their local training data using dataset distillation. Clients then use this compact distilled dataset during the unlearning and recovery phases rather than their full training datasets. This drastically reduces the computation cost of unlearning. The paper also proposes integrating dataset distillation into the regular federated learning training process by reusing gradients, avoiding extra overhead. Evaluations on MNIST, CIFAR-10, and SVHN show QuickDrop reduces unlearning time by 463.8x compared to full model retraining and 65.1x compared to prior FU techniques.

**Strengths:**

1. Proposes QuickDrop, which integrates dataset distillation into regular federated learning training by reusing gradients, avoiding extra overhead.
2. Experiments on 3 datasets conclusively show QuickDrop reduces unlearning time by 65-464x over baselines.

**Weaknesses:**

1. The paper does not discuss communication overhead of exchanging distilled datasets, which is the main concern I have regarding whether the proposed QuickDrop could be useful.
2. Some of the important details are not clear, for example, how non-IID data affects QuickDrop and the impact of distillation hyperparameters.

**Questions:**

I don't have other questions in addition to the questions in the weaknesses section.

---

> ### Author Response · Authors · 2023-11-15
> **Communication Overhead Issue**
>
> There may be a misunderstanding here.
>
> QuickDrop does not exchange distilled datasets: each distilled dataset is generated locally and remains local. Therefore, this should not be a concern at all. Unlearning, however, does proceed in a federated manner involving the exchange of updated and global models between clients and the server.
>
> We have provided in Section 3.2 a detailed explanation of the unlearning algorithm that might help clarify the point. If specific points still need to be clarified, we would be happy to provide additional details or discuss them further.

---

> ### Author Response · Authors · 2023-11-15
> **How non-IID data affects QuickDrop and the impact of distillation hyperparameters**
>
> **The effect of Non-IIDness on QuickDrop**:
>
> From an algorithm perspective, the data distribution of the dataset used does not impact how distilled data is generated on each client. From a performance perspective, in Appendix C (Table 7 and 8), we have conducted experiments with IID and non-IID data distributions, showing that QuickDrop can still achieve similar accuracy compared to baselines while maintaining its significant efficiency benefit.
>
> In general, non-IIDness affects us the same as in other unlearning algorithms/baselines. Understanding the impact of data distribution on unlearning algorithms and their proper assessment with well-defined metrics is still an open question for the machine unlearning community.
>
> **Impact of other hyperparameters for distillation**:
>
> **Scale**: An important hyperparameter that influences our distillation is the scale, which defines the size of the distilled dataset on each client. We currently use a scale of 100, i.e., we condense client local datasets by a hundred times. If we reduce the scale (e.g., to 10), it is obvious that we will obtain a larger distilled dataset, which can absorb more information to represent the characteristics of the original dataset and improve the effectiveness of unlearning.
>
> Given the importance of this parameter, we conduct additional experiments to test the influence of scale on unlearning accuracy and computation cost of QuickDrop (on CIFAR-10, alpha=0.1, 10 clients). We list the results in the tables below, where the first table shows the accuracy on the R-set after recovery and the second table shows the compute time to conduct unlearning:
>
> | Scale | Acc. on R-set  (QuickDrop) | Acc.on R-set  (Retraining) |
> |:-----:|:------------------------------:|:-------------------------------:|
> |  100  |             70.48%             |              74.95%             |
> |   10  |             73.69%             |              74.95%             |
>
> | Scale | Compute time  (QuickDrop) | Compute time (Retraining) |
> |:-----:|:-------------------------:|:-------------------------:|
> |  100  |           15.61s          |          7239.58s         |
> |   10  |          154.79s          |          7239.58s         |
>
> In conclusion, the scale is a trade-off between the unlearning accuracy and efficiency, and can be adjusted according to the different demands of clients or the FU environment. We will add these experimental results to the paper for a final version.
>
> We also identify other hyperparameters that influence the performance of QuickDrop, like the current global round K and local update steps T. In our work, we fix K=200 and T=50. Since QuickDrop combines DD and FL, the distilled dataset is a by-production of the normal FL training process. The local update step T and global round K required to obtain a trained FL model is much larger than the requirements of DD. Therefore, we are guaranteed that the distilled dataset with K=200 and T=50 are of high quality.

---

### Official Review · Reviewer_um5Z · 2023-10-30

**Soundness:** 3 good
**Presentation:** 2 fair
**Contribution:** 2 fair
**Rating:** 3
**Confidence:** 3

**Summary:**

This research is primarily focused on the development of a method designed for the removal of specific training data from a Machine Learning (ML) model that has been trained using Federated Learning (FL). This specialized technique, known as Federated Unlearning (FU), is the central objective of the study.

To achieve this, the authors introduce an innovative and highly efficient FU methodology named QUICKDROP. QUICKDROP leverages the power of Dataset Distillation (DD) to streamline the unlearning process, resulting in a significant reduction in the computational resources required when compared to conventional approaches. The fundamental principle of QUICKDROP involves each client generating a concise and representative dataset referred to as a "distilled dataset" using DD techniques. These distilled datasets play a crucial role in the subsequent unlearning phase.

Furthermore, the authors demonstrate their ingenuity by seamlessly integrating Dataset Distillation (DD) into the Federated Learning (FL) training process. This integration enhances the overall efficiency of QUICKDROP by capitalizing on the reuse of gradient updates generated during FL training for DD purposes. Consequently, the overhead associated with creating distilled datasets is effectively minimized.

The empirical evaluation of QUICKDROP's performance, conducted across three standard datasets, conclusively demonstrates its capacity to deliver remarkable efficiency gains in the context of federated unlearning.

**Strengths:**

1. The development of QUICKDROP significantly improves the efficiency of the Federated Unlearning (FU) process. By incorporating Dataset Distillation (DD) and Stochastic Gradient Ascent, the method considerably reduces the computational resources and time required for unlearning compared to existing approaches.

2. The research demonstrates the scalability of QUICKDROP by evaluating its performance with a large number of clients (100 clients). This scalability is vital in practical FL scenarios involving numerous participants.

**Weaknesses:**

The integration of Dataset Distillation (DD) and Federated Learning (FL) is not a novel concept. Previous research presented at conferences such as ICLR and CVPR has explored similar strategies involving the replacement of original data with distilled data for FL training. Applying this strategy to a new application problem, such as federated unlearning, does not represent a sufficiently novel contribution.

The experimental results presented in the research do not demonstrate strong evidence of effectiveness. While the findings indicate a significant improvement in efficiency, there are concerns about the effectiveness of the method, particularly in terms of accuracy. Sacrificing effectiveness for increased efficiency may not be a favorable trade-off in practical applications.

The organization of the paper could benefit from improvement. For instance, the extensive space dedicated to explaining dataset distillation through gradient matching is unnecessary. This concept is derived from existing work and is not a novel contribution of this research.

**Questions:**

Please refer to "Weaknesses" part.

---

> ### Author Response · Authors · 2023-11-15
> **Novelty about the integration of Dataset Distillation (DD) and Federated Learning (FL)**
>
> We acknowledge that Dataset Distillation (DD) within Federated Learning (FL) algorithms has been explored in prior research. However, our work introduces a meaningful extension by focusing on the application of DD to Federated Unlearning (FU), which presents distinct challenges compared to standard FL. The differences between FL and FU are non-trivial, and it is not immediately evident that DD performs well for FU. Additionally, we carefully reuse the gradients produced by FL to generate the distilled datasets, an improvement beyond simply applying DD to FU. Gradient reusage brings along a unique set of challenges. For instance, Zhao et al. (2021) train over several model initializations deemed necessary for good distillation, whereas only one model initialization is available for distillation in QuickDrop since we leveraged FL. We will elaborate on these challenges in the final version of our work.
>
> Our contributions also extend beyond the combination of existing methods in extensively demonstrating the viability, challenges, and modifications necessary to achieve efficient unlearning. We showcased this through thorough empirical assessments on three diverse datasets, each exhibiting varying levels of heterogeneity. Our work emphasizes minimizing the overhead of generating distilled datasets, addressing a critical aspect often overlooked in previous studies. While we agree that QuickDrop is inspired by and relies to some extent on existing techniques, our research is the first to apply DD to FU.

---

> ### Author Response · Authors · 2023-11-15
> **The effectiveness of QuickDrop on the trade-off between accuracy and efficiency.**
>
> We agree with the reviewer that our approach does not match the performance achieved by retraining from scratch baseline without additional fine-tuning steps. Like any methodology, inherent tradeoffs are inevitable, and it would be unrealistic to expect our method to consistently outperform retraining from scratch across all dimensions. We intend to present these results to offer a nuanced understanding of the efficiency-accuracy tradeoff.
>
> Even with a potential decrease in accuracy, the significant improvement in efficiency addresses a specific need in practical applications where computational resources might be limited (e.g., edge settings such as mobile phones). Furthermore, the efficiency gains of QuickDrop become more pronounced when doing multiple unlearning requests, as discussed in Section 4.3.
>
> We believe that our work contributes to this by providing a comprehensive picture of the tradeoff landscape, offering practitioers the flexibility to make informed decisions based on the specific requirements of their applications.

---

> ### Author Response · Authors · 2023-11-15
> **Paper Organization**
>
> Through the current organization, we intend to provide a comprehensive comparison of our proposed method to previous work, highlighting the key differences and modifications. Furthermore, this explanation is helpful for reviewers unaware of the inner workings of gradient matching. However, based on the reviewer’s feedback, we understand that this description might be too elaborate, and we intend to shorten this section in the final version.

---

### Public Comment · ~Saurav_Prakash1 · 2023-11-14
**Missing Related Work**

Dear authors,

Please consider citing our following related work:

Chao Pan, Jin Sima, Saurav Prakash, Vishal Rana, and Olgica Milenkovic. "Machine Unlearning of Federated Clusters." International Conference on Learning Representations (ICLR) 2023. Link to ICLR camera-ready version: https://openreview.net/forum?id=VzwfoFyYDga. Link to arXiv: https://arxiv.org/abs/2210.16424

Particularly, it is the first work that enables exact unlearning in the federated setting, and comes with provable theoretical guarantees on model performance during the unlearning process. Hence, we would appreciate it if you added a reference to our work.

Thanks,

Saurav

---

### Author Response · Authors · 2023-11-15
**General Response**

We thank the reviewers for their comments. Below, we provide several high-level comments that address concerns raised by multiple reviewers. We then provide a detailed rebuttal per reviewer.

**Novelty**: We notice that most reviewers question the novelty of our work as it appears as a combination of existing techniques. Our work proposes a novel and efficient approach to apply dataset distillation techniques to Machine Unlearning and, more specifically, Federated Unlearning. The concept of Federated Unlearning is becoming increasingly important when working with data-driven technologies. With the growing emphasis on data privacy and regulations like the General Data Protection Regulation (GDPR), it is essential to develop methods to efficiently remove individual data points from trained models without sharing raw data between clients. This need arises in various scenarios, such as users revoking consent for data usage, legal requirements for data deletion, or simply correcting data errors. While our QuickDrop algorithm reuses existing techniques, such as Dataset Distillation, it contributes to the efficiency of the unlearning process in FU settings and we are the first to apply DD in the context of FU.

**The locality of Dataset Distillation**: Each client performs Dataset Distillation locally during model training. No distilled datasets nor raw data points are exchanged between clients. Besides, the fine-tuning of each client’s distilled dataset is conducted independently on their local dataset, without any extra communication cost.

---

### Meta-Review · Area_Chair_koae · 2023-12-05

**Metareview:**

This paper uses dataset distillation for efficient machine unlearning (using stochastic gradient ascent on the distilled data).

The reviewers appreciated the basic idea; however, there were several concerns as well, such as lack of baselines (Reviewer ug1m), lack of theoretical guarantees (Reviewer BiDM), and lack of novelty (raised by several reviewers).

The authors' response was considered and discussed. However, the concerns still remained.

Based on all the reviews, the discussion, and my own reading, I agree with the reviewers that the paper in its current state is not strong enough to be published. The authors are advised to think about the concerns raised by the reviewers to improve the paper and consider resubmitting to another venue.

**Justification For Why Not Higher Score:**

The paper has several issues as pointed out by the reviewers and some of which are summarized in the meta-review as well.

**Justification For Why Not Lower Score:**

N/A

---

### Decision · Program_Chairs · 2024-01-16

Reject